

# Intraspecific variation in the diet of the Mexican garter snake *Thamnophis eques*

Javier Manjarrez[1], Martha Pacheco-Tinoco[1] and Crystian S. Venegas-Barrera[2]

[1] Facultad de Ciencias, Universidad Autónoma del Estado de México, Toluca, Estado de Mexico, Mexico
[2] División de Estudios de Posgrado e Investigación, Instituto Tecnológico de Ciudad Victoria, Ciudad Victoria, Tamaulipas, México

## ABSTRACT

The Mexican Garter Snake (*Thamnophis eques)* is a terrestrial-aquatic generalist that feeds on both aquatic and terrestrial prey. We describe size-related variation and sexual variation in the diet of *T. eques* through analysis of 262 samples of identifiable stomach contents in snakes from 23 locations on the Mexican Plateau. The snake *T. eques* we studied consumed mostly fish, followed in lesser amounts by leeches, earthworms, frogs, and tadpoles. Correspondence analysis suggested that the frequency of consumption of various prey items differed between the categories of age but not between sex of snakes, and the general pattern was a reduction of prey item diversity with size of snake. Snake length was correlated positively with mass of ingested prey. Large snakes consumed large prey and continued to consume smaller prey. In general, no differences were found between the prey taxa of male and female snakes, although males ate two times more tadpoles than females. Males and females did not differ in the mass of leeches, earthworms, fishes, frogs and tadpoles that they ate, and males and females that ate each prey taxon were similar in length. We discuss proximate and functional determinants of diet and suggest that the observed intraspecific variation in *T. eques* could be explored by temporal variation in prey availability, proportions of snake size classes and possible sexual dimorphism in head traits and prey dimensions to assess the role of intersexual resource competition.

## INTRODUCTION

The Mexican Garter Snake (*T. eques*) is a medium-sized garter snake classified as a generalist predator because it feeds on both aquatic and terrestrial prey; mostly frogs, tadpoles, and fish, supplemented by lizards and mice (*Drummond & Macías García, 1989*; *Manjarrez, 1998*). *Drummond & Macías García (1989)* found that *T. eques* at Tecocomulco, Hidalgo, is locally specialized in feeding on only two to three taxa. This snake forages in vegetative cover along the shore and an attack may include a sudden lunge across the surface toward prey (*Drummond & Macías García, 1989*).

Although *T. eques* is widely distributed over the Mexican Plateau, in this area, the disturbance and loss of habitat have caused the isolation and fragmentation of their populations (*Conant, 2003*; *Manjarrez, Contreras-Garduño & Janczur, 2014*), with low population densities and constricted distribution (*Rossman, Ford & Seigel, 1996*; *Manjarrez,*

Corresponding author
Javier Manjarrez,
jsilva@ecologia.unam.mx

*1998*). This scenario raises the possibility of intraspecific dietary differences by spatial variation of the environment. In general, garter snakes show important ecological intraspecific variation (*Rossman, Ford & Seigel, 1996*), and reports on diet for *T. eques* in Mexico showed sexual, ontogenetic (neonates-adults) and seasonal (rainy-dry) divergence in prey size (*Macías Garcia & Drummond, 1988*; *Drummond & Macías García, 1989*; *Manjarrez, Contreras-Garduño & Janczur, 2014*). For example, at Lake Tecocomulco, Mexico, small snakes fed mainly on aquatic invertebrates (leeches and earthworms), while large snakes fed on aquatic vertebrates (frogs, fish, and salamander larvae). Fluctuations in prey availability was associated with seasonal variation in prey (*Macías Garcia & Drummond, 1988*). At Toluca, Mexico, snakes *T. eques* were detected to have eaten earthworms, tadpole, slugs and mice (*Manjarrez, 1998*; *Manjarrez, Contreras-Garduño & Janczur, 2014*).

In this study, we provide the first broad description of the diet of *T. eques* on the Mexican Plateau. We looked for variation in consumption of prey type, sex and size-classes of snake from three different drainages. To permit more extensive and novel comparisons, we pooled our dietary records with those of *Lozoya (1988)* and *Drummond & Macías García (1989)*, as described in 'Materials and Methods'.

Sexual differences in snake diets show that females sometimes ingest larger prey than males (*Shine, 1993*; *Seigel, 1996*; *Daltry, Wuster & Thorpe, 1998*) and this difference is usually attributed to sexual dimorphism in body size when females are bigger that males. The maximum size of prey that can be ingested is constrained by a snake's gape (e.g., *King, 2002*), and in most species, larger snakes take larger prey and appear to drop small prey from their diet, although data from very young snakes is usually limited (review in *Arnold, 1993*). Garter snakes are sexually dimorphic in body size (*Shine, 1993*) and their diet can vary with age/size-classes (*Mushinsky, 1987*; *Macías Garcia & Drummond, 1988*; amongst others). Female garter snakes are usually larger than males (*Shine, 1994*) but sexual differences in garter snake diets have not been well studied (*Seigel, 1996*). *Thamnophis eques* is sexually dimorphic, with adult females being 5.6% larger than males in snout-vent length (SVL; *Manjarrez, 1998*; *Manjarrez, Contreras-Garduño & Janczur, 2014*).

## MATERIALS & METHODS

This study received the approval of the ethics committee of the Universidad Autónoma del Estado de México (Number 4047/2016SF). All subjects were treated humanely on the basis of guidelines outlined by the American Society of Ichthyologists and Herpetologists (*ASIH, 2004*).

We collected snakes along streams, rivers, canals, ponds, and lakes. We measured each captured snake (SVL in cm), and also recorded sex in adults by visual inspection of the tail-base breadth or by manually everting the hemipenis in small snakes (*Conant & Collins, 1998*). Although the snakes were weighed, these data were not used in this study. Each snake was forced to regurgitate stomach contents by abdominal palpation (*Carpenter, 1952*). After processing, snakes were released at their capture sites. We measured the wet mass of each prey item and then fixed them in 10% formalin and preserved them in 70% alcohol.

We sampled snake stomach contents at 23 sites on the Mexican Plateau (Lerma, Tula, and Nazas, drainages, Table S1) sporadically during the active reproductive season (February–November) over a period of 16 years (Table S2). The sites in the Lerma and Tula drainages were sampled from 1980 to 1986 and 1991 to 1995. We obtained 194 regurgitations from 22 sites in Jalisco, Michoacán, México, Hidalgo, and Queretaro. The records obtained for these two drainages are partially reported by *Lozoya* (*1988*, 19% of the total regurgitations), a reference inaccessible to most readers.

In the Nazas drainage we obtained 68 regurgitations. The Nazas population inhabits an isolated 0.36 ha spring-fed cattle pond in the Chihuahuan Desert with a rainy season from June through October. The records were obtained during 27 2–4 day visits, bimonthly during April to November 1981, and monthly during February through December 1982, and February through November 1983. *Drummond & Macías García (1989)* previously reported 13% of the total these records. Snakes were captured on the first two days of each visit and released on the second day to prevent repeat sampling during the same visit. We counted and classified prey items as fish, leeches, frogs, tadpoles, and earthworms.

## Analysis

We classified snakes as neonates (<20.5 cm SVL), juveniles (20.5–39.5 cm) or adults (>39.5 cm; *Manjarrez, 1998*). Percentages of regurgitations containing each prey taxon were normalized by arcsine transformation (*Zar, 1984*). We used MANCOVA (with snake length as a covariate) to explore variation in the mass of prey consumed by prey taxa and by the sex of each snake. We included complementary analyses of dietary variation in relation to snake size whenever these could contribute to understanding variation in diet. For analyses of prey mass we excluded taxa represented by fewer than five prey items. Prey mass and snake length were natural log transformed prior to calculating correlations because of the lack of homoscedasticity and skewed distributions.

To avoid making Type I or II errors when many Chi square ($X^2$) tests are performed, we used a Multiple Correspondence Analysis (MCA) to associate prey items consumed more frequently (earthworm, tadpole, fish, frog and leech, considered as not continuous data) by the combination of two categorical variables, the sex of snake (male–female) and snake size (neonates, juveniles, and adults). The analysis is a modification of $X^2$ used to analyze contingency tables and creates a Cartesian diagram based on the association between more than three categorical variables (*Legendre & Legendre, 2003*). The diagram display simultaneously the relative position (canonical position) of studied variables categories (*Gotelli, 2001*). The nearest canonical positions of different variables represent a high association, while distant categories show lower association. The degrees of freedom (*df*) and probability (*P*) for MCA have no statistical relevance, because both were used only in two-way table; therefore, we do not report these values. However, similar to $X^2$ test, the MCA estimate the differences between observed and expected values, which allow estimate contribution of each variable to $X^2$ test value. The analyses generates a coordinate system of reference that account variations of all variables (dimensions), where we reported the first two dimensions, which represents the higher variations. Finally, the center of graph

(coordinate 0, 0) was the average of all variables, therefore categories nearest to center of graph show a lower association to rest of categories.

Additionally, we performed a Cluster analysis to identify groups of categories of age and sex of snakes with similar consumption of prey, we used the Morisita index of similarity between frequencies of prey consumed, and the Ward's Method of amalgamation (*Rencher, 2002*). Ward's Method attempts to minimize the Sum of Squares (SS) of any two hypothetical clusters that can be formed at each step (*Legendre & Legendre, 2003*), and it is considered as very efficient method, due reduce cluster long chains produced by other joining amalgamation methods as nearest neighbor or UPGMA. The number of groups retained were determined by graph of amalgamation schedule. The graph shows a line graph of the linkage distances at successive clustering steps, the optimal cut-off to deciding how many clusters to retain is when the linkage distance line forms a plateau. Multiple Correspondence and Cluster Analyses are complementary, the MCA identify associations among categories of different categorical variables, but cannot determine differences; while cluster create groups of different elements, but we cannot know why they are different.

## RESULTS

### Prey items

We obtained identifiable stomach contents from 262 (38%) of 690 *T. eques* collected. Thirteen regurgitations (5.7% of total) included more than one prey species, and hence contributed more than one data point for some snakes. In order of percentage of prey items identified we found 42.4% ($n = 111$ regurgitations) consumed fish, 23.7% ($n = 62$) leeches, 10.6% ($n = 28$) earthworms, 10.2% ($n = 27$) frogs, 9.8% ($n = 25$) tadpoles, and 2.3% ($n = 6$) axolotls (*Ambystoma sp.*). Lizard, slug, and mouse were recovered from only 1 stomach each.

Our Multiple Correspondence Analysis ($X^2 = 2{,}410.1$) shows that prey type variable contributes with 71.6% of $X^2$ test value ($X^2 = 1{,}725.9$), while snake size with 19.5% ($X^2 = 468.6$) and sex with 8.9% ($X^2 = 215.6$). Canonical position of adult snakes was closer to tadpoles, frogs, and axolotls, therefore adults tend to feed more frequently these prey types than juveniles and neonates snakes (Fig. 1). Juveniles consumed more earthworms, while neonates principally leeches (Fig. 1). Canonical position of male and female snakes showed a lower variation on the prey type consumed (Fig. 1). Cluster analysis indicated that the three size classes differ in prey type consumed (Fig. 2), where adult snakes were closer to juveniles, and both distant to neonate. Prey types consumed by female and male snakes were similar, due linkage distance nodes of males and females of the same size classes was lower than cutoff value. This grouping of snakes suggests a scheme of ontogenetic change in the taxon of prey, with lower relevance of the grouping by sex (Fig. 2).

### Variation with snake length

Snakes of different sizes ate a changing diversity of prey types. The general pattern was a reduction of number of prey item with increasing snake body size. (Fig. 3). Snakes <65 cm SVL ate all prey types of all sizes, including invertebrates (leeches and earthworms) and vertebrates (tadpoles, fish, and frogs). Vertebrate prey were taken by only the largest snakes
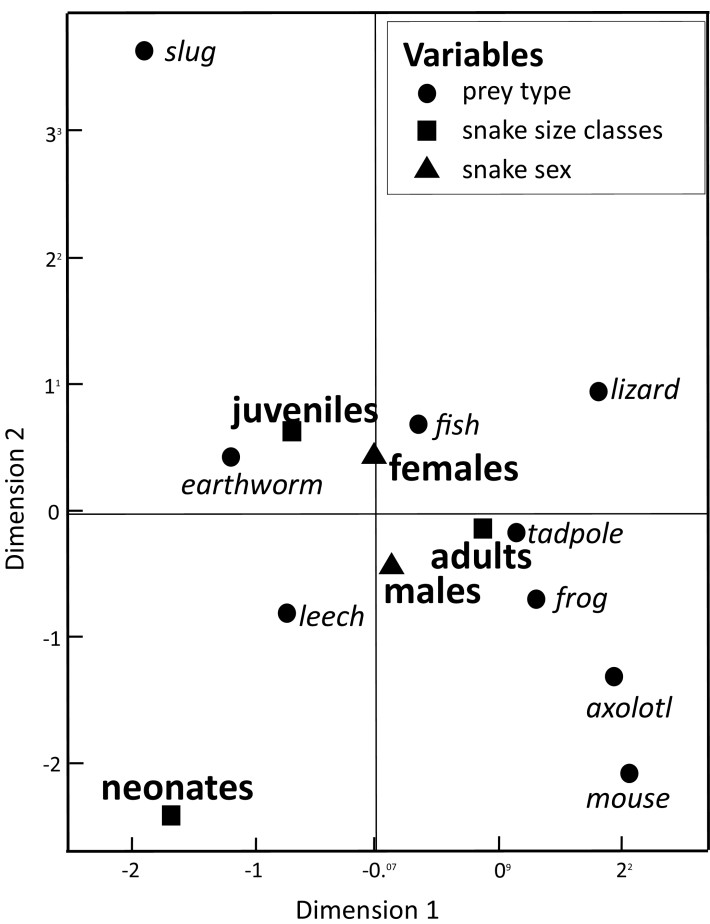

**Figure 1** Dimension variables obtained from a Multiple Correspondence Analysis to associate prey items consumed by snake *T. eques* in the combination of snake sex (male–female) and snake size classes (neonates, juveniles, and adults).

(>60 cm SVL). At 15 cm SVL, snakes eat leeches as one of two major prey items, but at 55 cm SVL consumption of leeches decreases drastically and disappears completely in larger snakes >65 cm SVL (Fig. 3). The consumption of fish and tadpoles increases when snake body size increases. However, the longer snakes >75 cm SVL consume only fish and tadpoles while excluding all other prey, possibly because longer snakes were a very small part of the entire sample ($n = 7$ stomach contents).

Snake length was correlated positively with mass of ingested prey ($r = 0.42$, $F_{1,\,326} = 71.52$, $P < 0.001$; Fig. 4). Large snakes consume large prey and continue to consume smaller prey. The same relationship was presented for leeches ($r = 0.42$, $F_{1,\,136} = 29.85$, $P < 0.00$; Fig. 5) and fish ($r = 0.43$, $F_{1,\,88} = 20.30$, $P < 0.001$; Fig. 5), but not correlated with earthworm ($r = 0.14$, $F_{1,\,32} = 18.32$, $P = 0.806$; Fig. 5) and tadpole mass ($r = 0.2$, $F_{1,\,25} = 2.36$, $P = 0.136$; Fig. 5).

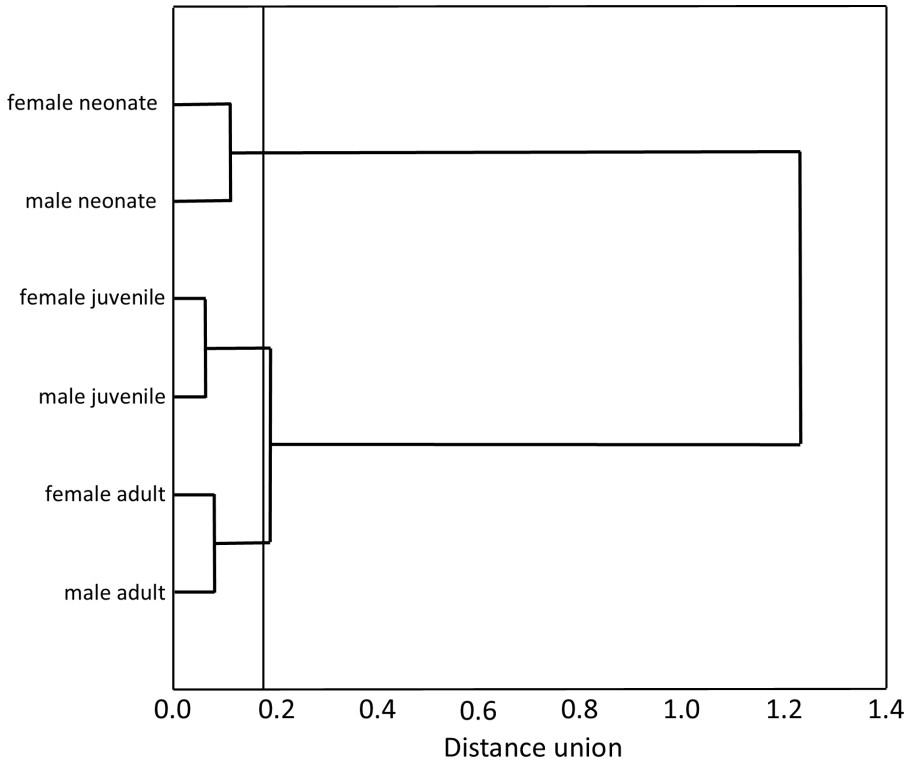

**Figure 2** **Hierarchical tree produced agglomeration of size classes (neonates, juveniles and adults) and sex (female and male) of *T. eques* snakes in function of prey type consumed.** The amalgamation schedule (Ward's Method) defined in 0.8 the cutoff value for the tree diagram.

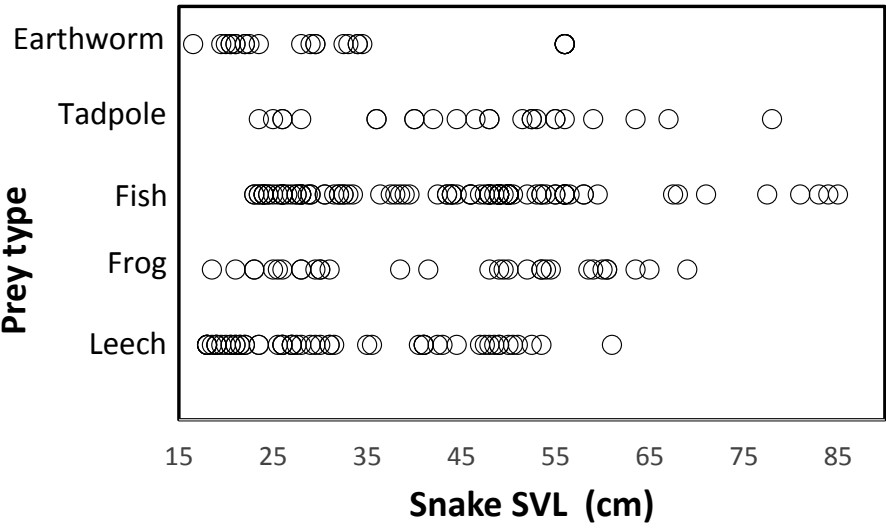

**Figure 3** **Relationship between prey type and snake size (SVL, cm) of *T. eques* in México.** Each circle represents an individual snake with a type of prey item (262 regurgitations).

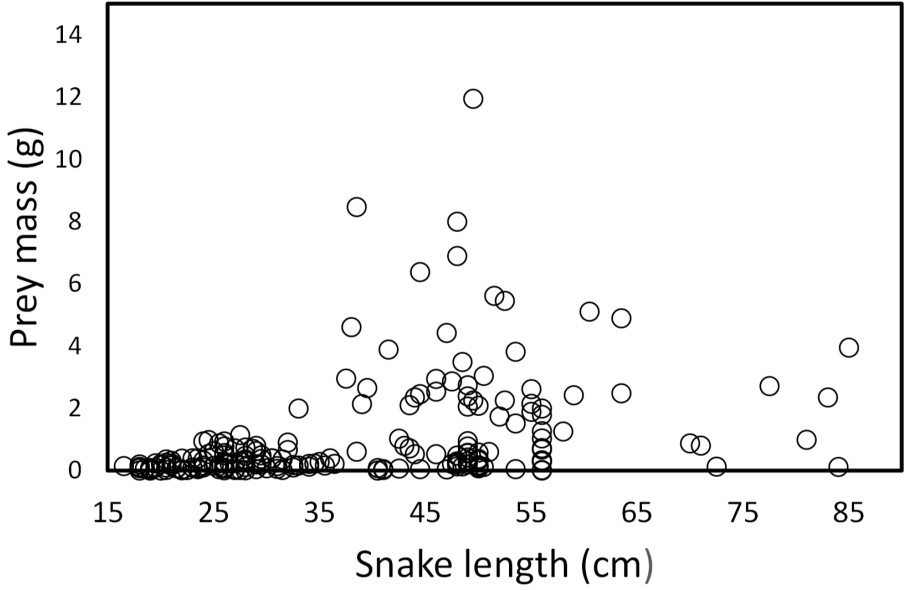

**Figure 4** **Prey mass as a function of snake length (SVL, cm) of (*T. eques*) in México ($r = 0.42$, $F_{1, 326} = 71.52$, $P < 0.001$).** Each dot represents an individual prey item. The many circles on the zero line of prey mass are because the low weights of leeches and earthworms.

### Variation with snake sex

Mean body lengths of captured snakes did not differ between sexes of neonates (Student $t_{20} = 0.08$, $P = 0.94$), juveniles (Student $t_{94} = 0.95$, $P = 0.34$), or adults (Student $t_{131} = 0.13$, $P = 0.90$); thus male and female snakes were similar in size.

No differences were found between the diets of male and female snakes. Pooling all sizes of snake, males ($n = 124$) and females ($n = 121$) ate similar proportions of the five main prey taxa ($X^2 = 3.82$, $P = 0.43$), both sexes eating mainly fishes, frogs, leeches, and earthworms, and in similar proportions. Males ate two times more tadpoles (0.13) than females (0.06).

Males and females did not differ in the mass of leeches, earthworms, fishes, frogs and tadpoles that they ate (MANCOVA $F_{1, 193} = 0.79$, $P = 0.37$), and males and females that ate each prey taxon were similar in length ($F_{1, 235} = 0.91$, $P = 0.34$).

## DISCUSSION

In this study, we provide a broad description of the diet of *T. eques* on the Mexican Plateau. The results indicate 69% of total regurgitations contain two major prey: leeches and fish, while the other three main prey are ingested in similar percentages (earthworms, 10.6%; frogs, 10.2%; tadpoles, 9.8%). The diet of *T. eques* included amphibious prey (frogs), terrestrial prey (earthworms) and aquatic prey (leeches, fish and tadpoles), and occasionally other prey such as axolotls, slugs, lizards, and mice. The main prey include three vertebrates (65%) and two invertebrates (35%).
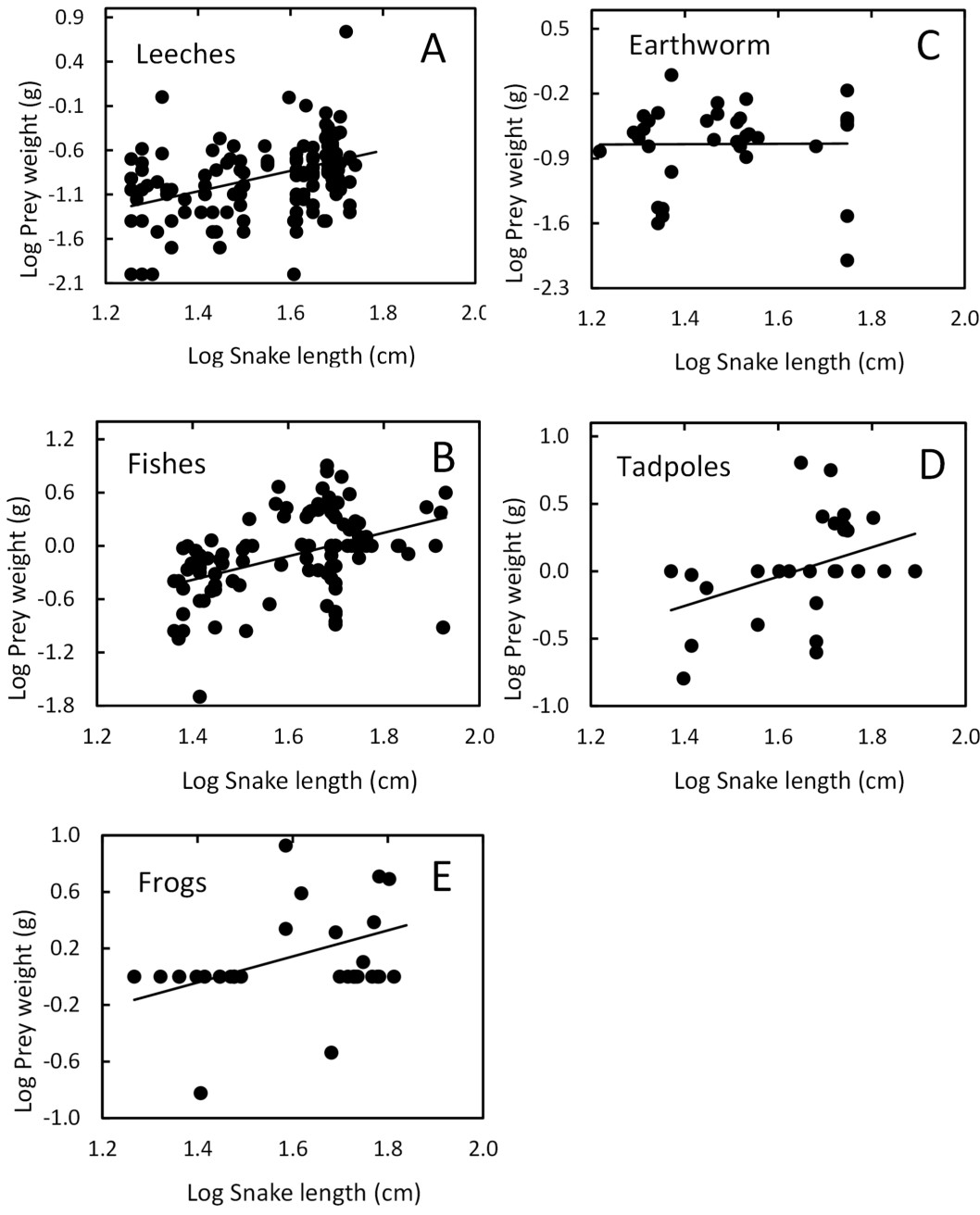

**Figure 5** **Relation between prey mass and snake length of *T. eques*.** (A) Leeches ($r = 0.42$, $P < 0.00$), (B) fish ($r = 0.43$, $P < 0.001$), (C) earthworm ($r = 0.14$, $P = 0.806$), (D) tadpole ($r = 0.2$, $P = 0.136$), (E) frogs ($r = 0.597$, $P < 0.001$).

The studies that have analyzed the diet of *T. eques* in Mexico included four local descriptions and in each study *T. eques* is locally specialized in feeding on only 2–3 prey taxa (Table 1) (*Drummond & Macías García, 1989*). This suggests a pattern of spatial variation in the diet of *T. eques*, presumably by the local availability of prey, for example

**Table 1  Percentage of prey reported in the diet of *Thamnophis eques* in Mexico.**

| | Tecocomulco[a] 126 snakes | Cerrillo[b] 18 regurgitations | Toluca Valley[c] 148 snakes | Fresnillo[d] 64 snakes | Present study 262 regurgitations |
|---|---|---|---|---|---|
| **Invertebrate prey** | | | | | |
| Earthworm | 41 | 22 | 20.2 | 2.9 | 10.6 (*Eisenia foetida* and *Eisenia* sp.) |
| Leech | 39 | – | 8.7 | – | 23.7 (*Erpobdella punctata* and *Mooreobdella* sp.) |
| Slug | 1.0 | 5.5 | – | – | 0.4 |
| **Vertebrate prey** | | | | | |
| Fish | 11 | – | 29.0 | – | 42.4 (*Girardinichthys multiradiatus*, *Carassius auratus*) |
| Frog | 5 | 28 | 10.1 | 69.0 | 10.2 (*Rana berlandieri* and *Rana* sp.) |
| Tadpole | 1.5 | 33 | 22.2 | 23.4 | 9.8 (*R. berlandieri*, *Rana* sp.) |
| Other (axolotl, lizard or mouse) | 4.5 | 11.0 | 9.4 | 4.7 | 3.2 |

**Notes.**
[a] *Macías Garcia & Drummond (1988)*.
[b] *Manjarrez (1998)*.
[c] *Manjarrez, Contreras-Garduño & Janczur (2014)*.
[d] *Drummond & Macías García (1989)*.

the temporal variation of prey, which has not yet been explored for *T. eques* (*Gregory & Nelson, 1991*; *Tuttle & Gregory, 2009*).

The diet of *T. eques* can also exhibit ontogenetic variations associated with individual size, changing from terrestrial to aquatic prey as snake size increases (*Macías Garcia & Drummond, 1988*; *Drummond & Macías García, 1989*). Ontogenetic change by prey taxa in gartersnakes, could be attributed to proximate mechanisms such as morphological constraints that determine the size of ingested prey (*Shine, 1991*; *Arnold, 1993*), the availability of potential prey (*Krebs, 2009*), energy or nutritional needs (*Britt, Hicks & Bennett, 2006*), habituation and learning (*Halloy & Burghardt, 1990*; *Ford & Burghardt, 1993*) or genetically programmed preferences (*Arnold, 1977*; *Arnold, 1981*; *Britt, Hicks & Bennett, 2006*). In Tecocomulco, Hidalgo, the differential distribution of large and small snakes was interpreted as a possible cause of differences in diet of *T. eques* with differences in the pattern of foraging, so that the snake can be an effective predator in the air-water interface; preying on aquatic prey when they are particularly vulnerable and terrestrial prey being added to the diet only opportunistically (*Drummond & Macías García, 1989*).

In our study, the ontogenetic variation in diet of *T. eques* was also found in the relationship between snake size and prey mass. The ingested prey size gradually increases with snake size and large snakes continued eating small prey (*Arnold, 1993*). This relationship could be interpreted as an ontogenetic telescope (*Arnold, 1993*), as previously reported for *T. eques* in a Zacatecas population with *Rana berlandieri* (*Drummond & Macías García, 1989*).

The absence of ontogenetic variation in the regurgitated samples of tadpoles and earthworms in *T. eques*, is common because in a previous study the annelids were ingested by *T. eques* regardless of snake body size (*Macías Garcia & Drummond, 1988*). The proximate explanation for this phenomenon is the high availability of these prey

during the annual active period of the snake or by a stable ontogenetic preference for invertebrates (*Ford & Burghardt, 1993*). The proximate and functional diet determinants of intraspecific variation in *T. eques* could be explored by local and temporal variation in prey availability and proportions of snake size classes collected. Intraspecific differences in *Thamnophis* diet have been described for different populations, seasons, years, size classes, and sexes (*Rossman, Ford & Seigel, 1996*). For example, in some species of *Thamnophis* have been described geographic dietary differences (e.g., *T. elegans*, *T. radix*, and *T. sirtalis*; *Kephart, 1982*; *Kephart & Arnold, 1982*; *Tuttle & Gregory, 2009*) explained by spatial or temporal variation in prey availability (*Kephart & Arnold, 1982*; *Seigel, 1996*).

The intersexual variation in food habits has been associated with sexual differences in body size (*Shine et al., 1998*). *Thamnophis eques* has been reported as sexually size dimorphic, with males smaller than females in SVL (*Manjarrez, 1998*); however in this study, the average size of male and female *T. eques* were similar and there were no sexual differences in diet, except that males ate two times more tadpoles than females. This sexual difference in diet can probably be attributed to real diet preference because females and males that ate each prey taxon were similar in length. Generally, large samples revealed no differences between male and female snakes in variety of prey taxa, proportions of different prey taxa and taxon specific prey weight (*Shine, 1993*), and the males and females that ate each taxon were similar in size. Overall, male and female *T. eques* differ in size (*Manjarrez, Contreras-Garduño & Janczur, 2014*), microhabitat use (*Venegas-Barrera, 2001*), seasonal foraging pattern (*Drummond & Macías García, 1989*), and diet (*Macías Garcia & Drummond, 1988*; *Manjarrez, Contreras-Garduño & Janczur, 2014*). As was found, males and females of this species do not differ in the body size of prey and type of prey consumed. The possible small differences in diet and microhabitat can be expose by larger sample sizes. Prey size and energetic demands may determine developmental transitions to different prey sizes or taxa, whereas sex, in this snake lacking sexual size dimorphism, has little or no influence on diet. However, sexual dimorphism in head dimensions, and eaten prey shape have seldom been searched and it will be essential to measure prey and head traits for *T. eques* to evaluate the function of resource competition between sexes.

## CONCLUSIONS

In this study, we provide the first broad description of the diet of the snake *T. eques* on the Mexican Plateau. The two major prey were leeches and fish. The diet of *T. eques* included amphibious, terrestrial and aquatic prey with ontogenetic variations associated with individual size, changing from terrestrial to aquatic prey as snake size increases. The ontogenetic variation in diet of *T. eques* was also found in the relationship between snake size and prey mass. The average size of male and female *T. eques* were similar and there were no sexual differences in diet. The proximate and functional diet determinants of intraspecific variation in *T. eques* could be explored by local and temporal variation in prey availability and proportions of snake size classes collected.

## ACKNOWLEDGEMENTS

For their assistance in the field and laboratory work we thank Hugh Drummond, Constantino Macías García and all of the students of the Evolutionary Biology Laboratory. We also thank Ruthe J. Smith for her comments and corrections of the typescript. All subjects were treated humanely on the basis of guidelines outlined by the Society for the Study of Amphibians & Reptiles. C Zepeda, M Manjarrez-Zepeda, and J Manjarrez-Zepeda provided moral support to JM.

### Funding

This study was supported by a research grant from the Universidad Autónoma del Estado de México (3589/2013SF, 4047/2016SF). The funders had no role in study design, data collection and analysis, decision to publish, or preparation of the manuscript.

### Grant Disclosures

The following grant information was disclosed by the authors:
Universidad Autónoma del Estado de México: 3589/2013SF, 4047/2016SF.

### Competing Interests

The authors declare there are no competing interests.

### Author Contributions

- Javier Manjarrez conceived and designed the experiments, performed the experiments, wrote the paper, prepared figures and/or tables, reviewed drafts of the paper.
- Martha Pacheco-Tinoco performed the experiments, analyzed the data, prepared figures and/or tables, collected data.
- Crystian S. Venegas-Barrera analyzed the data, contributed reagents/materials/analysis tools, prepared figures and/or tables, reviewed drafts of the paper.

### Animal Ethics

The following information was supplied relating to ethical approvals (i.e., approving body and any reference numbers):

This study received the approval of the ethics committee of the Universidad Autónoma del Estado de México (Number 4047/2016SF). All subjects were treated humanely on the basis of guidelines outlined by the American Society of Ichthyologists and Herpetologists (*ASIH, 2004*).

### Data Availability

The raw data has been supplied as Data S1.

### Supplemental Information

Supplemental information for this article can be found online at http://dx.doi.org/10.7717/peerj.4036#supplemental-information.

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
