# Peer review of "Intraspecific variation in the diet of the Mexican garter snake Thamnophis eques"

_PeerJ, doi:10.7717/peerj.4036_

## Round 0.1 · original submission · Major Revisions

Please revise your manuscript according to the reviewers' concerns. Be advised that the revised version will be returned to the reviewers, so make sure that you carefully address all of their criticisms.

·

Basic reporting

This is a beautiful data set that covers many years. Well done. The statistical relevance needs to be included in the legends of each figure. I would suggest attempting to combine the final figure of log snake length and prey mass into one chart. the use of different symbols or colors would be appropriate, however, this may prove to be too messy and unreadable.

Experimental design

I applaud this manuscript as there are not enough natural history style manuscripts being written. My sole comment in this section is why were the hemipenes or cloaca not everted to determine sex? Simply eyeballing the tail breadth will invariably lead to misidentifying the sex of some animals.

Validity of the findings

I would be careful on commenting on the commonness of some of your findings. You have stated your conclusions well and described the diet of these populations clearly. I have added some questions in the text, such as what is the percentage of snakes that had food? I also encourage you to be wary of digestion level of food. I believe that this has led to many circles on the zero line of figure 3. I would consider adding this information to the legend or text, or both.

Additional comments

This is a robust data set that you have done a nice job with. I have identified some concerns in the text and in the notes. One thing to add is to include a citation for your delineations of snake size classifications. Have those size classifications been used before, are you classifying them as newborn, etc.. upon personal observation, natural history notes, etc...

Reviewer 2 ·

Basic reporting

The authors have presented a valuable contribution on the diet of T. eques, a relatively little studied gartersnake. They attempt to describe diet differences between sexes and among size classes, which is an important aspect of the species ecology.

This MS needs to have major editing for grammar and language. There are portions that are difficult to follow because of grammar issues.

Authors need to follow consistent format for the species name. As is they switch between scientific and common name.

Authors need to follow proper and consistent format for presenting stats.

I made edits, comments, and suggestions to the PDF. These address questions that need to be clarified and statements that need more justification.

Experimental design

The overall design is simple and appropriate.

I made questions and comments on the statistical approaches that I would like to see addressed.

In the introduction that authors state they pooling older data (Lines 62-64) in their analyses, but it is unclear if this was actually done. As the methods and results currently read the authors only deal with their data. I would like to see them conduct tests with their data, that of the other authors, and the pooled data. Alternatively, instead of doing this independently, they can do a multivariate test using the data source as a variable to determine if diet varied among the data sources. Either way this issue needs to be addressed and clarified.

The authors need to clarify the sex body size analyses because it is unclear to me if this analysis was done with only adults or included all snakes (juveniles and adults). Including all snakes may bias their results to no sexual size dimorphism.

Validity of the findings

As currently presented the results are novel by contributing to our knowledge of the snakes diet between age classes, but I am not completely convinced about the sex differences. See above comment and those to the MS.

I made additional comments to the MS regarding the methods.

Many of the questions and comments made to the discussion may be addressed if other sections are more clearly explained. I found it difficult to connect methods, results, and discussion because of the lack of detail in the methods. Sometimes it is hard to follow the logic and methods the authors use, making further interpretations difficult.

Additional comments

This MS has a lot of potential and the authors have presented valuable data. But in its current state I feel there is more work to do, especially in regards to more clearly explaining the methods and results.

For ontogenetic diet and prey availability discussion I suggest the authors look into the rich body of literature on other Thamnophis species.

Annotated reviews are not available for download in order to protect the identity of reviewers who chose to remain anonymous.

---

## Round 0.2 · Minor Revisions

The reviewers have noted a number of additional revisions which would greatly improve the manuscript, and I concur. Please address these additional concerns in another revision.

·

Basic reporting

There has been considerable improvement in many areas of the MS. However, it reads more disjunct with this improvements. Certain sections suffer from poor grammar while other sections flow very well. I found it difficult to jump from common name to scientific name throughout the MS. Please identify the scientific and stick with it. Line 71, the citations listed are not complete. i would suggest adding the phrase 'amongst others'. In addition, Line 73 the author identifies that recent studies do not investigate sexual diff in diet. i find this to be4 inaccurate and deserves a more comprehensive literature search to address this. The final paragraph of the introduction is entirely out of place.

Experimental design

The question is appropriate and approached correctly. Line 82: snakes were collected haphazardly at 3 different locales on the Mexican plateau. the reader does not need to know about flipping rocks, etc.. Line 92: 16 years on 1.5 +- 0.9 occasions each, what is being referred to here? Line 101 13% of these findings, again, what is being referred to here?

Validity of the findings

The additional analysis performed, the MCA contributes to the robustness of the data. It demonstrates the data can be analyzed in multiple ways to produce the same results. I find the MCA analysis much more informative than the cluster analysis and would recommend using that in supplemental materials. The analysis specifically describes slugs and axolotls on the figures, however, table 1 simply classifies these as others. It would be best to stay congruent with these data. Line 233-237 grammar is not present.Line 221 citations are needed because this has been found in numerous other snake species.

Additional comments

I found your improvements and grammar and additional analysis evident. However, the MS suffers from being very disjunct at this point, to the reader. I feel like some sections are beautifully written and others are grammatically lacking. I would suggest a couple of more citations (as noted above). I think the new analysis adds to the robustness of the data. I prefer it greatly to the cluster. Please include examples of the "other" section of table 1 as you address slugs and axolotl specifically in Fig 1. overall good improvements.

Reviewer 2 ·

Basic reporting

I am pleased with the authors revisions, especially with how the addressed the statistical issues. The authors have done a great job on this manuscript. I made a couple of very minor edits which are noted on the attached PDF.

Experimental design

Addressed concerns.

Validity of the findings

Addressed concerns.

Additional comments

Good job!

Annotated reviews are not available for download in order to protect the identity of reviewers who chose to remain anonymous.

---

## Round 0.3 · accepted · Accept

Thank you very much for your revisions.